# Neural Bridge Sampling for Evaluating Safety-Critical Autonomous Systems

**Aman Sinha**[*]
Stanford University
amans@stanford.edu

**Matthew O'Kelly**[*]
University of Pennsylvania
mokelly@seas.upenn.edu

**Russ Tedrake**
Massachusetts Institute of Technology
russt@mit.edu

**John Duchi**
Stanford University
jduchi@stanford.edu

## Abstract

Learning-based methodologies increasingly find applications in safety-critical domains like autonomous driving and medical robotics. Due to the rare nature of dangerous events, real-world testing is prohibitively expensive and unscalable. In this work, we employ a probabilistic approach to safety evaluation in simulation, where we are concerned with computing the probability of dangerous events. We develop a novel rare-event simulation method that combines exploration, exploitation, and optimization techniques to find failure modes and estimate their rate of occurrence. We provide rigorous guarantees for the performance of our method in terms of both statistical and computational efficiency. Finally, we demonstrate the efficacy of our approach on a variety of scenarios, illustrating its usefulness as a tool for rapid sensitivity analysis and model comparison that are essential to developing and testing safety-critical autonomous systems.

## 1   Introduction

Data-driven and learning-based approaches have the potential to enable robots and autonomous systems that intelligently interact with unstructured environments. Unfortunately, evaluating the performance of the closed-loop system is challenging, limiting the success of such methods in safety-critical settings. Even if we produce a deep reinforcement learning agent better than a human at driving, flying a plane, or performing surgery, we have no tractable way to certify the system's quality. Thus, currently deployed safety-critical autonomous systems are limited to structured environments that allow mechanisms such as PID control, simple verifiable protocols, or convex optimization to enable guarantees for properties like stability, consensus, or recursive feasibility (see *e.g.* [33, 69, 14]). The stylized settings of these problems and the limited expressivity of guaranteeable properties are barriers to solving unstructured, real-world tasks such as autonomous navigation, locomotion, and manipulation.

The goal of this paper is to *efficiently* evaluate complex systems that lack safety guarantees and/or operate in unstructured environments. We assume access to a simulator to test the system's performance. Given a distribution $X \sim P_0$ of simulation parameters that describe typical environments for the system under test, our governing problem is to estimate the probability of an adverse event

$$p_\gamma := \mathbb{P}_0(f(X) \leq \gamma). \tag{1}$$

The parameter $\gamma$ is a threshold defining an adverse event, and $f : \mathcal{X} \to \mathbb{R}$ measures the safety of a realization $x$ of the agent and environment (higher values are safer). In this work, we assume $P_0$ is

---

[*]Equal contribution

known; the system-identification and generative-modeling literatures (*e.g.* [6, 82]) provide several approaches to learn or specify $P_0$. A major challenge for solving problem (1) is that the better an agent is at performing a task (*i.e.* the smaller $p_\gamma$ is), the harder it is to confidently estimate $p_\gamma$—one rarely observes events with $f(x) \leq \gamma$. For example, when $P_0$ is light-tailed, the sample complexity of estimating $p_\gamma$ using naive Monte Carlo samples grows exponentially [19].

Problem (1) is often solved in practice by naive Monte Carlo estimation methods, the simplest of which *explore* the search space via random samples from $P_0$. These methods are unbiased and easy to parallelize, but they exhibit poor sample complexity. Naive Monte Carlo can be improved by adding an adaptive component *exploiting* the most informative portions of random samples drawn from a sequence of approximating distributions $P_0, P_1, \ldots, P_K$. However, standard adaptive Monte Carlo methods (*e.g.* [20]), though they may use first-order information on the distributions $P_k$ themselves, fail to use first-order information about $f$ to improve sampling; we explicitly leverage this to accelerate convergence of the estimate through *optimization*.

Naive applications of first-order optimization methods in the estimation problem (1)—for example biasing a sample in the direction $-\nabla f(x)$ to decrease $f(x)$—also require second-order information to correct for the distortion of measure that such transformations induce. Consider the change of variables formula for distributions $\rho(y) = \rho(g^{-1}(y)) \cdot |\det J_{g^{-1}}(y)|$ where $y = g(x)$. When $g(x)$ is a function of the gradient $\nabla f(x)$, the volume distortion $|\det J_{g^{-1}}(y)|$ is a function of the Hessian $\nabla^2 f(x)$. Hessian computation, if even defined, is unacceptably expensive for high-dimensional spaces $\mathcal{X}$ and/or simulations that involve the time-evolution of a dynamical system; our approach avoids any Hessian computation. In contrast, gradients $\nabla f(x)$ can be efficiently computed for many closed-loop systems [1, 80, 107, 59] or through the use of surrogate methods [105, 28, 36, 8].

To that end, we propose *neural bridge sampling*, a technique that combines *exploration, exploitation*, and *optimization* to efficiently solve the estimation problem (1). Specifically, we consider a novel Markov-chain Monte Carlo (MCMC) scheme that moves along an adaptive ladder of intermediate distributions $P_k$ (with corresponding unnormalized densities $\rho_k(x)$ and normalizing constants $Z_k := \int_{\mathcal{X}} \rho_k(x) dx$). This MCMC scheme iteratively transforms the base distribution $P_0$ to the distribution of interest $P_0 I\{f(x) \leq \gamma\}$. Neural bridge sampling adaptively balances exploration in the search space (via $\nabla \log \rho_0$) against optimization (via $\nabla f$), while avoiding Hessian computations. Our final estimate $\hat{p}_\gamma$ is a function of the ratios $Z_k / Z_{k-1}$ of the intermediate distributions $P_k$, the so-called "bridges" [10, 66]. We accurately estimate these ratios by warping the space between the distributions $P_k$ using neural density estimation.

**Contributions and outline**    Section 2 presents our method, while Section 3 provides guarantees for its statistical performance and overall efficiency. A major focus of this work is empirical, and accordingly, Section 4 empirically demonstrates the superiority of neural bridge sampling over competing techniques in a variety of applications: (i) we evaluate the sensitivity of a formally-verified system to domain shift, (ii) we consider design optimization for high-precision rockets, and (iii) we perform model comparisons for two learning-based approaches to autonomous navigation.

## 1.1   Related Work

**Safety evaluation**    Several communities [27] have attempted to evaluate the closed-loop performance of cyber-physical, robotic, and embodied agents both with and without learning-based components. Existing solutions are predicated on the definition of the evaluation problem: verification, falsification, or estimation. In this paper we consider a method that utilizes interactions with a gradient oracle in order to solve the estimation problem (1). In contrast to our approach, the verification community has developed tools (*e.g.* [56, 24, 4]) to investigate whether any adverse or unsafe executions of the system exist. Such methods can certify that failures are impossible, but they require that the model is written in a formal language (a barrier for realistic systems), and they require whitebox access to this formal model. Falsification approaches (*e.g.* [40, 31, 5, 108, 34, 83]) attempt to find *any* failure cases for the system (but not the overall probability of failure). Similar to our approach, some falsification approaches (*e.g.* [1, 107]) utilize gradient information, but their goal is to simply minimize $f(x)$ rather than solve problem (1). Adversarial machine learning is closely related to falsification; the key difference is the domain over which the search for falsifying evidence is conducted. Adversarial examples (*e.g.* [61, 53, 95, 99]) are typically restricted to a $p$-norm ball around a point from a dataset, whereas falsification considers all possible in-distribution examples.

Both verification and falsification methods provide less information about the system under test than estimation-based methods: they return only whether or not the system satisfies a specification. When the system operates in an unstructured environment (*e.g.* driving in an urban setting), the mere existence of failures is trivial to demonstrate [93]. Several authors (*e.g.* [76, 104]) have proposed that it is more important in such settings to understand the overall frequency of failures as well as the relative likelihoods of different failure modes, motivating our approach.

**Sampling techniques and density estimation**    When sampling rare events and estimating their probability, there are two main branches of related work: parametric adaptive importance sampling (AIS) [63, 75] and nonparametric sequential Monte Carlo (SMC) techniques [32, 30]. Both of these literatures are advanced forms of variance reduction techniques, and they are complementary to standard methods such as control variates [91, 46]. Parametric AIS techniques, such as the cross-entropy method [90], postulate a family of distributions for the optimal importance-sampling distribution. They iteratively perform heuristic optimization procedures to update the sampling distribution. SMC techniques perform sampling from a sequence of probability distributions defined nonparametrically by the samples themselves. The SMC formalism encompasses particle filters, birth-death processes, and smoothing filters [29]. Our technique blends aspects of both of these communities: we include parametric warping distributions in the form of normalizing flows [82] within the SMC setting.

Our method employs bridge sampling [10, 66], which is closely related to other SMC techniques such as umbrella sampling [23], multilevel splitting [16, 20], and path sampling [41]. The operational difference between these methods is in the form of the intermediate distribution used to calculate the ratio of normalizing constants. Namely, the optimal umbrella sampling distribution is more brittle than that of bridge sampling [23]. Multilevel splitting employs hard barriers through indicator functions, whereas our approach relaxes these hard barriers with smoother exponential barriers. Path sampling generalizes bridge sampling by taking discrete bridges to a continuous limit; this approach is difficult to implement in an adaptive fashion.

The accuracy of bridge sampling depends on the overlap between intermediate distributions $P_k$. Simply increasing the number of intermediate distributions is inefficient, because it requires running more simulations. Instead, we employ a technique known as *warping*, where we map intermediate distributions to a common reference distribution [102, 65]. Specifically, we use normalizing flows [86, 54, 81, 82], which efficiently transform arbitrary distributions to standard Gaussians through a series of deterministic, invertible functions. Normalizing flows are typically used for probabilistic modeling, variational inference, and representation learning. Recently, Hoffman et al. [47] explored the benefits of using normalizing flows for reparametrizing distributions within MCMC; our warping technique encompasses this benefit and extends it to the SMC setting.

**Beyond simulation**    This paper assumes that the generative model $P_0$ of the operating domain is given, so all failures are in the modeled domain by definition. When deploying systems in the real world, anomaly detection [22] can discover distribution shifts and is complementary to our approach (see *e.g.* [26, 68]). Alternatively, the problem of distribution shift can be addressed offline via distributional robustness [39, 70, 84], where we analyze the worst-case probability of failure under an uncertainty set composed of perturbations to $P_0$.

## 2    Proposed approach

As we note in Section 1, naive Monte Carlo measures probabilities of rare events inefficiently. Instead, we consider a sequential Monte Carlo (SMC) approach: we decompose the rare-event probability $p_\gamma$ into a chain of intermediate quantities, each of which is tractable to compute with standard Monte Carlo methods. Specifically, consider $K$ distributions $P_k$ with corresponding (unnormalized) probability densities $\rho_k$ and normalizing constants $Z_k := \int_{\mathcal{X}} \rho_k(x)dx$. Let $\rho_0$ correspond to the density for $P_0$ and $\rho_\infty(x) := \rho_0(x)I\{f(x) \leq \gamma\}$ be the (unnormalized) conditional density for the region of interest. Then, we consider the following decomposition:

$$p_\gamma := \mathbb{P}_0(f(X) \leq \gamma) = \mathbb{E}_{P_K}\left[\frac{Z_K}{Z_0}\frac{\rho_\infty(X)}{\rho_K(X)}\right], \qquad \frac{Z_K}{Z_0} = \prod_{k=1}^{K}\frac{Z_k}{Z_{k-1}}. \qquad (2)$$

**Algorithm 1** Neural bridge sampling

---

**Input:** $N$ samples $x_i^0 \overset{\text{i.i.d.}}{\sim} P_0$, MCMC steps $T$, step size $\alpha \in (0,1)$, stop condition $s \in (0,1)$
Initialize $k \leftarrow 0, \beta_0 \leftarrow 0, \log(\hat{p}_\gamma) \leftarrow 0$
**while** $\frac{1}{N} \sum_i I\{f(x_i^k) \leq \gamma\} < s$ **do**
  $\beta_{k+1} \leftarrow$ solve problem (8)
  **for** $i = 1$ to $N$, in parallel
    $x_i^{k+1} \overset{\text{i.i.d.}}{\sim} \text{Mult}(\{\rho_{k+1}(x_i^k)/\rho_k(x_i^k)\})$  // multinomial resampling
  **for** $t = 1$ to $T$
    **for** $i = 1$ to $N$, in parallel
      $x_i^{k+1} \leftarrow \text{WarpedHMC}(x_i^k, \theta_k)$  // Appendix A
  $\theta_{k+1} \leftarrow$ argmin problem (6)  // train normalizing flow on $\{x_i^{k+1}\}$ via SGD
  $\log(\hat{p}_\gamma) \leftarrow \log(\hat{p}_\gamma) + \log(Z_{k+1}/Z_k)$  // warped bridge estimate (5)
  $k \leftarrow k + 1$
$\log(\hat{p}_\gamma) \leftarrow \log(\hat{p}_\gamma) + \log(\frac{1}{N} \sum_i I\{f(x_i^k) \leq \gamma\})$

---

Although we are free to choose the intermediate distributions arbitrarily, we will show below that our estimate for each ratio $Z_k/Z_{k-1}$ and thus $p_\gamma$ is accurate insofar as the distributions sufficiently overlap (a concept we make rigorous in Section 3). Thus, the intermediate distributions act as bridges that iteratively steer samples from $P_0$ towards $P_K$. One special case is the multilevel splitting approach [50, 16, 104, 74], where $\rho_k(x) := \rho_0(x)I\{f(x) \leq L_k\}$ for levels $\infty =: L_0 > L_1 \ldots > L_K := \gamma$. In this paper, we introduce an exponential tilting barrier [94]

$$\rho_k(x) := \rho_0(x) \exp\left(\beta_k \left[\gamma - f(x)\right]_-\right), \tag{3}$$

which allows us to take advantage of gradients $\nabla f(x)$. Here we use the "negative ReLU" function defined as $[x]_- := -[-x]_+ = xI\{x < 0\}$, and we assume that the measure of non-differentiable points, *e.g.* where $\nabla f(x)$ does not exist or $f(x) = \gamma$, is zero (see Appendix A for a detailed discussion of this assumption). We set $\beta_0 := 0$ and adaptively choose $\beta_k > \beta_{k-1}$. The parameter $\beta_k$ tilts the distribution towards the distribution of interest: $\rho_k \to \rho_\infty$ as $\beta_k \to \infty$. In what follows, we describe an MCMC method that combines exploration, exploitation, and optimization to draw samples $X_i^k \sim P_k$. We then show how to compute the ratios $Z_k/Z_{k-1}$ given samples from both $P_{k-1}$ and $P_k$. Finally, we describe an adaptive way to choose the intermediate distributions $P_k$. Algorithm 1 summarizes the overall approach.

**MCMC with an exponential barrier**    Gradient-based MCMC techniques such as the Metropolis-adjusted Langevin algorithm (MALA) [89, 88] or Hamiltonian Monte Carlo (HMC) [35, 73] use gradients $\nabla \log \rho_0(x)$ to efficiently explore the space $\mathcal{X}$ and avoid inefficient random-walk behavior [37, 25]. Classical mechanics inspires the HMC approach: HMC introduces an auxiliary random momentum variable $v \in \mathcal{V}$ and generates proposals by performing Hamiltonian dynamics in the augmented state-space $\mathcal{X} \times \mathcal{V}$. These dynamics conserve volume in the augmented state-space, even when performed with discrete time steps [58].

By including the barrier $\exp\left(\beta_k \left[\gamma - f(x)\right]_-\right)$, we combine exploration with optimization; the magnitude of $\beta_k$ in the barrier modulates the importance of $\nabla f$ (optimization) over $\nabla \log \rho_0$ (exploration), two elements of the HMC proposal (see Appendix A for details). We discuss the adaptive choice for $\beta_k$ below. Most importantly, we avoid any need for Hessian computation because the dynamics conserve volume. As Algorithm 1 shows, we perform MCMC as follows: given $N$ samples $x_i^{k-1} \sim P_{k-1}$ and a threshold $\beta_k$, we first resample using their importance weights (exploiting the performance of samples that have lower function value than others) and then perform $T$ HMC steps. In this paper, we implement split HMC [92] which is convenient for dealing with the decomposition of $\log \rho_k(x)$ into $\log \rho_0(x) + \beta_k[\gamma - f(x)]_-$ (see Appendix A for details).

**Estimating $Z_k/Z_{k-1}$ via bridge sampling**    Bridge sampling [10, 66] allows estimating the ratio of normalizing constants of two distributions by rewriting

$$E_k := \frac{Z_k}{Z_{k-1}} = \frac{Z_k^B/Z_{k-1}}{Z_k^B/Z_k} = \frac{\mathbb{E}_{P_{k-1}}[\rho_k^B(X)/\rho_{k-1}(X)]}{\mathbb{E}_{P_k}[\rho_k^B(X)/\rho_k(X)]}, \qquad \widehat{E}_k = \frac{\sum_{i=1}^N \rho_k^B(x_i^{k-1})/\rho_{k-1}(x_i^{k-1})}{\sum_{i=1}^N \rho_k^B(x_i^k)/\rho_k(x_i^k)}, \tag{4}$$

where $\rho_k^B$ is the density for a bridge distribution between $P_{k-1}$ and $P_k$, and $Z_k^B$ is its associated normalizing constant. We employ the geometric bridge $\rho_k^B(x) := \sqrt{\rho_{k-1}(x)\rho_k(x)}$. In addition to

being simple to compute, bridge sampling with a geometric bridge enjoys the asymptotic performance guarantee that the relative mean-square error scales inversely with the Bhattacharyya coefficient, $G(P_{k-1}, P_k) = \int_{\mathcal{X}} \sqrt{\frac{\rho_{k-1}(x)}{Z_{k-1}} \frac{\rho_k(x)}{Z_k}} dx \in [0, 1]$ (see Appendix B for a proof). This value is closely related to the Hellinger distance, $H(P_{k-1}, P_k) = \sqrt{2 - 2G(P_{k-1}, P_k)}$. In Section 3, we analyze the ramifications of this fact on the overall convergence of our method.

**Neural warping**    Both HMC and bridge sampling benefit from warping samples $x_i$ into a different space. As Betancourt [11] notes, HMC mixes poorly in spaces with ill-conditioned geometries. Girolami and Calderhead [42] and Hoffman et al. [47] explore techniques to improve mixing efficiency by minimizing shear in the corresponding Hamiltonian dynamics. One way to do so is to transform to a space that resembles a standard isotropic Gaussian [62].

Conveniently, transforming $P_k$ to a common distribution (*e.g.* a standard Gaussian) also benefits the bridge-sampling estimator (4). As noted above, the error of the bridge estimator grows with the Hellinger distance between the distributions $H(P_{k-1}, P_k)$. However, normalizing constants $Z_k$ are invariant to (invertible) transformations. Thus, transformations that warp the space between distributions reduce the error of the bridge-sampling estimator (4). Concretely, we consider invertible transformations $W_k$ such that $y_i^k = W_k(x_i^k)$. For clarity of notation, we write probability densities over the space $\mathcal{Y}$ as $\phi$, the corresponding distributions for $Y^k$ as $Q_k$, and the the inverse transformations $W_k^{-1}(y)$ as $V_k(y)$. Then we can write the bridge-sampling estimate (4) in terms of the transformed variables $y$. The numerator and denominator are as follows:

$$\mathbb{E}_{Q_{k-1}}\left[\frac{\phi_k^B(Y)}{\phi_{k-1}(Y)}\right] = \mathbb{E}_{Q_{k-1}}\left[\sqrt{\frac{\phi_k(Y)}{\phi_{k-1}(Y)}}\right] = \mathbb{E}_{Q_{k-1}}\left[\sqrt{\frac{\rho_k(V_k(Y))|\det J_{V_k}(Y)|}{\rho_{k-1}(V_{k-1}(Y))|\det J_{V_{k-1}}(Y)|}}\right], \quad (5a)$$

$$\mathbb{E}_{Q_k}\left[\frac{\phi_k^B(Y)}{\phi_k(Y)}\right] = \mathbb{E}_{Q_k}\left[\sqrt{\frac{\phi_{k-1}(Y)}{\phi_k(Y)}}\right] = \mathbb{E}_{Q_k}\left[\sqrt{\frac{\rho_{k-1}(V_{k-1}(Y))|\det J_{V_{k-1}}(Y)|}{\rho_k(V_k(Y))|\det J_{V_k}(Y)|}}\right]. \quad (5b)$$

By transforming all $P_k$ into $Q_k$ to resemble standard Gaussians, we reduce the Hellinger distance $H(Q_{k-1}, Q_k) \leq H(P_{k-1}, P_k)$. Note that the volume distortions in the expression (5) are functions of the transformation $V_k$, so they do not require computation of the Hessian $\nabla^2 f$. However, computing $\rho_k(V_k(y))$ requires evaluations of $f$ (*e.g.* calls of the simulator). We consider the cost-benefit analysis of warping in Section 3.

Classical warping techniques include simple mean shifts or affine scaling [102, 65]. Similar to Hoffman et al. [47], we consider normalizing flows, a much more expressive class of transformations that have efficient Jacobian computations [82]. Specifically, given samples $x_i^k$, we train masked autoregressive flows (MAFs) [81] to minimize the empirical KL divergence between the transformed samples $y_i^k$ and a standard Gaussian $D_{\mathrm{KL}}(Q_k \| \mathcal{N}(0, I))$. Parametrizing $W_k$ by $\theta_k$, this minimization problem is equivalent to:

$$\text{minimize}_\theta \sum_{i=1}^{N} -\log\left|\det J_{W_k}\left(x_i^k; \theta\right)\right| + \frac{1}{2}\left\|W_k\left(x_i^k; \theta\right)\right\|_2^2. \quad (6)$$

The KL divergence is an upper bound to the Hellinger distance; we found minimizing the former to be more stable than minimizing the latter. Furthermore, to improve training efficiency, we exploit the iterated nature of the problem and warm-start the weights $\theta_k$ with the trained values $\theta_{k-1}$ when solving problem (6) via stochastic gradient descent (SGD). As a side benefit, the trained flows can be repurposed as importance-samplers for the ladder of distributions from nominal behavior to failure.

**Adaptive intermediate distributions**    Because we assume no prior knowledge of the system under test, we exploit previous progress to choose the intermediate $\beta_k$ online; this is a key difference to our approach compared to other forms of sequential Monte Carlo (*e.g.* [71, 72]) which require a predetermined schedule for $\beta_k$. We define the quantities

$$a_k := \sum_i^N I\{f(x_i^k) \leq \gamma\}/N, \ \ b_k(\beta) := \sum_{i=1}^N \exp\left((\beta - \beta_k)[\gamma - f(x_i^k)]_-\right)/N. \quad (7)$$

The first is the fraction of samples that have achieved the threshold. The second is an importance-sampling estimate of $E_{k+1}$ given samples $x_i^k \sim P_k$, written as a function of $\beta$. For fixed fractions $\alpha, s \in (0, 1)$ with $\alpha < s$, $\beta_{k+1}$ solves the following optimization problem:

$$\text{maximize } \beta \ \text{s.t.} \ \{b_k(\beta) \geq \alpha, \ a_k/b_k(\beta) \leq s\}. \quad (8)$$

Since $b_k(\beta)$ is monotonically decreasing and $b_k(\beta) \geq a_k$, this problem can be solved efficiently via binary search. The constant $\alpha$ tunes how quickly we enter the tails of $P_0$ (smaller $\alpha$ means fewer iterations), whereas $s$ is a stop condition for the last iteration. Choosing $\beta_{k+1}$ via (8) yields a crude estimate for the ratio $Z_{k+1}/Z_k$ as $\alpha$ (or $a_{K-1}/s$ for the last iteration). The bridge-sampling estimate $\widehat{E}_{k+1}$ corrects this crude estimate once we have samples from the next distribution $P_{k+1}$.

## 3    Performance analysis

We can write the empirical estimator of the function (2) as

$$\hat{p}_\gamma = \prod_{k=1}^{K} \widehat{E}_k \frac{1}{N} \sum_{i=1}^{N} \frac{\rho_\infty(x_i^K)}{\rho_K(x_i^K)}, \tag{9}$$

where $\widehat{E}_k$ is given by the expression (4) without warping, or similarly, as a Monte Carlo estimate of the expression (5) with warping. We provide guarantees for both the time complexity of running Algorithm 1 (*i.e.* the iterations $K$) as well as the overall mean-square error of $\hat{p}_\gamma$. For simplicity, we provide results for the asymptotic (large $N$) and well-mixed MCMC (large $T$) limits. Assuming these conditions, we have the following:

**Proposition 1.** *Let $K_0 := \lfloor \log(p_\gamma)/\log(\alpha) \rfloor$. Then, for large $N$ and $T$, $s \geq 1/3$, and $p_\gamma < s$, the total number of iterations in Algorithm 1 approaches $K \overset{\text{a.s.}}{\to} K_0 + I\{p_\gamma/\alpha^{K_0} < s\}$. Furthermore, for the non-warped estimator, the asymptotic relative mean-square error $\mathbb{E}[(\hat{p}_\gamma/p_\gamma - 1)^2]$ is*

$$\frac{2}{N} \sum_{k=1}^{K} \left( \frac{1}{G(P_{k-1}, P_k)^2} - 1 \right) - \frac{2}{N} \sum_{k=1}^{K-1} \left( \frac{G(P_{k-1}, P_{k+1})}{G(P_{k-1}, P_k)G(P_k, P_{k+1})} - 1 \right) + \frac{1-s}{sN} + o\left( \frac{1}{N} \right). \tag{10}$$

*In particular, if the inverse Bhattacharyya coefficients are bounded such that $\frac{1}{G(P_{k-1}, P_k)^2} \leq D$ (with $D \geq 1$), then the asymptotic relative mean-square error satisfies $\mathbb{E}[(\hat{p}_\gamma/p_\gamma - 1)^2] \leq 2KD/N$. For the warped estimator, replace $G(P_i, P_j)$ with $G(Q_i, Q_j)$ in the expression (10).*

See Appendix B for the proof. We provide some remarks about the above result. Intuitively, the first term in the bound (10) accounts for the variance of $\widehat{E}_k$. The denominator of $\widehat{E}_{k-1}$ and numerator of $\widehat{E}_k$ both depend on $x_i^k$; the second sum in (10) accounts for the covariance between those terms. Furthermore, the quantities in the bound (10) are all empirically estimable, so we can compute the mean-square error from a single pass of Algorithm 1. In particular,

$$G(P_{k-1}, P_k)^2 = \frac{Z_k^B}{Z_{k-1}} \frac{Z_k^B}{Z_k}, \qquad \frac{G(P_{k-1}, P_{k+1})}{G(P_{k-1}, P_k)G(P_k, P_{k+1})} = \frac{Z_k^C}{Z_k} \frac{Z_k}{Z_k^B} \frac{Z_k}{Z_{k+1}^B}, \tag{11}$$

where $Z_k^C/Z_k = \mathbb{E}_{P_k}\left[ \rho_k^B(X)\rho_{k+1}^B(X)/\rho_k(X)^2 \right]$. The last term in the bound (10) is the relative variance of the final Monte Carlo estimate $\sum_i I\{f(x_i^K) \leq \gamma\}/N$.

**Overall efficiency**    The statistical efficiency outlined in Proposition 1 is pointless if it is accompanied by an overwhelming computational cost. We take the atomic unit of computation to be a query of the simulator, which returns both evaluations of $f(x)$ and $\nabla f(x)$; we assume other computations to be negligible compared to simulation. As such, the cost of Algorithm 1 is $N(1 + KT)$ evaluations of the simulator without warping and $N(1 + KT) + 2KN$ with warping. Thus, the relative burden of warping is minimal, because training the normalizing flows to minimize $D_{\text{KL}}(Q_k \| \mathcal{N}(0, I))$ requires no extra simulations. In contrast, directly minimizing $D_{\text{KL}}(Q_{k-1} \| Q_k)$ would require extra simulations at each training step to evaluate $\rho_k(V_k(y))$.

Our method can exploit two further sources of efficiency. First, we can employ surrogate models for gradient computation and/or function evaluation during the $T$ MCMC steps. For example, using a surrogate model for a fraction $d \leq 1 - 1/T$ of the MCMC iterations reduces the factor $T$ to $T_s := (1 - d)T$ in the overall cost. Surrogate models have an added benefit of making our approach amenable for simulators that do not provide gradients. The second source of efficiency is parallel computation. Given $C$ processors, the factor $N$ in the cost drops to $N_c := \lceil N/C \rceil$.

The overall efficiency of the estimator (9)—relative error multiplied by cost [44]—depends on $p_\gamma$ as $\log(p_\gamma)^2$. In contrast, the standard Monte Carlo estimator has cost $N$ to produce an estimate with

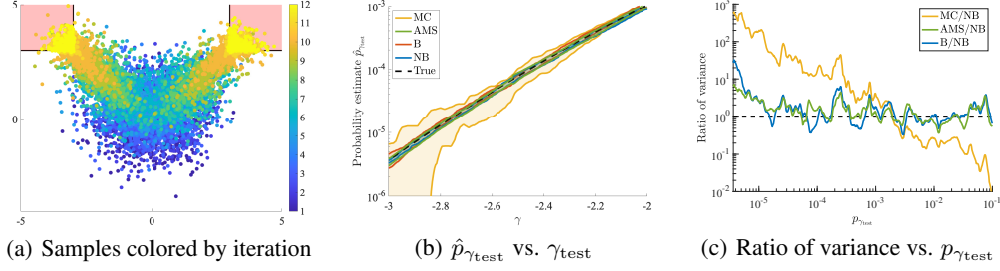

(a) Samples colored by iteration     (b) $\hat{p}_{\gamma_{\text{test}}}$ vs. $\gamma_{\text{test}}$     (c) Ratio of variance vs. $p_{\gamma_{\text{test}}}$

**Figure 1.** Experiments on a synthetic problem. 10 trials are used to calculate the 99% confidence intervals in (b) and variance ratios in (c). All adaptive methods perform similarly in this well-conditioned search space except at very small $\gamma$, where NB performs the best.

relative error $\frac{1-p_\gamma}{p_\gamma N}$. Thus, the relative efficiency gain for our estimator (9) over naive Monte Carlo is $O(1/(p_\gamma \log(p_\gamma)^2))$: the efficiency gains over naive Monte Carlo increase as $p_\gamma$ decreases.

## 4   Experiments

We evaluate our approach in a variety of scenarios, showcasing its use in efficiently evaluating the safety of autonomous systems. We begin with a synthetic problem to illustrate the methodology concretely as well as highlight the pitfalls of using gradients naively. Then, we evaluate a formally-verified neural network controller [48] on the OpenAI Gym continuous MountainCar environment [67, 17] under a domain perturbation. Finally, we consider two examples of using neural bridge sampling as a tool for engineering design in high-dimensional settings: (a) comparing thruster sizes to safely land a rocket [13] in the presence of wind, and (b) comparing two algorithms on the OpenAI Gym CarRacing environment (which requires a surrogate model for gradients) [55].

We compare our method with naive Monte Carlo (MC) and perform ablation studies for the effects of neural warping (denoted as NB with warping and B without). We also provide comparisons with adaptive multilevel splitting (AMS) [16, 104, 74]. All methods are given the same computational budget as measured by evaluations of the simulator. This varies from 50,000-100,000 queries to run Algorithm 1 as determined by $p_\gamma$ (see Appendix C for details of each experiment's hyperparameters). However, despite running Algorithm 1 with a given $\gamma$, we evaluate estimates $\hat{p}_{\gamma_{\text{test}}}$ for all $\gamma_{\text{test}} \geq \gamma$. Larger $\gamma_{\text{test}}$ require fewer queries to evaluate $\hat{p}_{\gamma_{\text{test}}}$ (as Algorithm 1 terminates early). Thus, we adjust the number of MC queries accordingly for each $\gamma_{\text{test}}$. Independently, we calculate the ground-truth values $p_{\gamma_{\text{test}}}$ for the non-synthetic problems using a fixed, very large number of MC queries.

**Synthetic problem**   We consider the two-dimensional function $f(x) = -\min(|x_{[1]}|, x_{[2]})$, where $x_{[i]}$ is the $i^{\text{th}}$ dimension of $x \in \mathbb{R}^2$. We let $\gamma = -3$ and $P_0 = \mathcal{N}(0, I)$ (for which $p_\gamma = 3.6 \cdot 10^{-6}$). Note that $\nabla^2 f(x) = 0$ almost everywhere, yet $\nabla f(x)$ has negative divergence in the neighborhoods of $x_{[2]} = |x_{[1]}|$. Indeed, gradient descent collapses $x_i \sim P_0$ to the lines $x_{[2]} = |x_{[1]}|$, and the ill-defined nature of the Hessian makes it unsuitable to track volume distortions. Thus, simple gradient-based transformations used to find adversarial examples (*e.g.* minimize $f(x)$) should not be used for estimation in the presence of non-smooth functions, unless volume distortions can be quantified.

Figure 1(a) shows the region of interest in pink and illustrates the gradual warping of $\rho_0$ towards $\rho_\infty$ over iterations of Algorithm 1. Figures 1(b) and 1(c) indicate that all adaptive methods outperform MC for $p_{\gamma_{\text{test}}} < 10^{-3}$. For larger $p_{\gamma_{\text{test}}}$, the overhead of the adaptive methods renders MC more efficient (Figure 1(c)). The linear trend of the yellow MC/NB line in Figure 1(c) aligns with the theoretical efficiency gain discussed in Section 3. Finally, due to the simplicity of the search space and the landscape of $f(x)$, the benefits of gradients and warping are not drastic. Specifically, as shown in Figure 1(c), all adaptive methods have similar confidence in their estimates except at very small $p_{\gamma_{\text{test}}} < 10^{-5}$, where NB outperforms AMS and B. The next example showcases the benefits of gradients as well as neural warping in a more complicated search space.

**Sensitivity of a formally-verified controller under domain perturbation**   We consider a minimal reinforcement learning task, the MountainCar problem [67] (Figure 2(a)). Ivanov et al. [48] created a formally-verified neural network controller to achieve reward $> 90$ over all initial positions

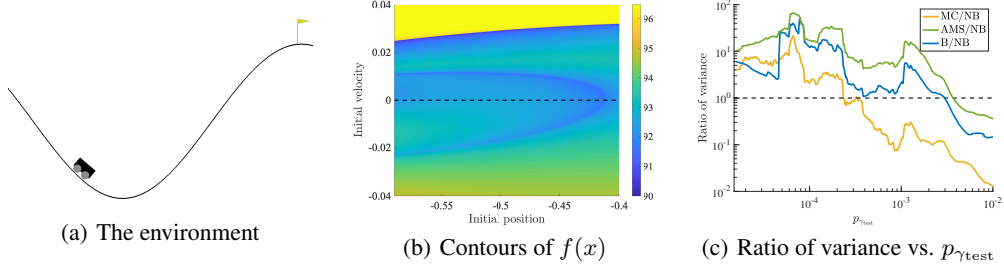

(a) The environment

(b) Contours of $f(x)$

(c) Ratio of variance vs. $p_{\gamma_{\text{test}}}$

**Figure 2.** Experiments on the MountainCar environment. The dashed horizontal line in (b) is the line along which the controller is formally verified. 10 trials are used for the variance ratios in (c). The irregular geometry degrades performance of AMS and B, but B benefits slightly from gradients over AMS. NB uses gradients and neural warping to outperform all other techniques.

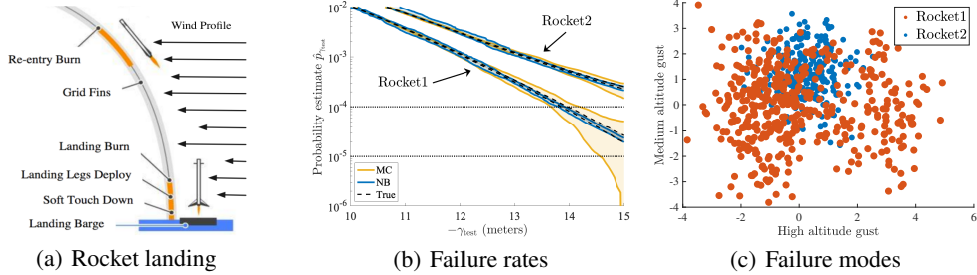

(a) Rocket landing

(b) Failure rates

(c) Failure modes

**Figure 3.** Rocket design experiments. NB's high-confidence estimates enable quick design iterations to either increase the landing pad radius or consider a third rocket that fails with probability $< 10^{-5}$. Low-dimensional visualization shows that Rocket2's failure types are more concentrated than those of Rocket1, even though Rocket2 has a higher overall probability of failure.

$\in [-0.59, -0.4]$ and 0 initial velocity (see Appendix C). The guarantees of formal verification hold only with respect to the specified domain; even small domain perturbations can affect system performance [49]. We illustrate this sensitivity by adding a small perturbation to the initial velocity $\sim \mathcal{N}(0, 10^{-4})$ and seek $p_\gamma := \mathbb{P}_0(\text{reward} \leq 90)$ for $P_0 = \text{Unif}(-0.59, -0.4) \times \mathcal{N}(0, 10^{-4})$. We measure the ground-truth failure rate as $p_\gamma = 1.6 \cdot 10^{-5}$ using 50 million naive Monte Carlo samples.

Figure 2(b) shows contours of $f(x)$. Notably, the failure region (dark blue) is an extremely irregular geometry with pathological curvature, which renders MCMC difficult for AMS and B [11]. Quantitatively, poor mixing adversely affects the performance of AMS and B, and they perform even worse than MC (Figure 2(c)). Whereas gradients help B slightly over AMS, gradients and neural warping together help NB outperform all other methods. We next move to higher-dimensional systems.

**Rocket design**   We now consider the problem of autonomous, high-precision vertical landing of an orbital-class rocket (Figure 3(a)), a technology first demonstrated by SpaceX in 2015. Rigorous system-evaluation techniques such as our risk-based framework are powerful tools for quickly exploring design tradeoffs. In this experiment, the amount of thrust which the rocket is capable of deploying to land safely must be balanced against the payload it is able to carry to space; stronger thrust increases safety but decreases payloads. We consider two rocket designs and we evaluate their respective probabilities of failure (not landing safely on the landing pad) for landing pad sizes up to 15 meters in radius. That is, $-f(x)$ is the distance from the landing pad's center at touchdown and $\gamma = -15$. We evaluate whether the rockets perform better than a threshold failure rate of $10^{-5}$.

We let $P_0$ be the 100-dimensional search space parametrizing the sequence of wind-gusts during the rocket's flight. Appendix C contains details for this parametrization and the closed-loop simulation of the rocket's control law (based on industry-standard approaches [13, 87]). Figure 3(b) shows the estimated performance of the two rockets. We show only MC and NB for clarity; comparisons with other methods are in Table 1 (with ground-truth values calculated using 50 million naive Monte Carlo simulations). Whereas both NB and MC confidently estimate Rocket2's failure rate as higher than $10^{-4}$, only NB confidently estimates Rocket1's failure rate as higher than $10^{-5}$, letting engineers quickly judge whether to increase the size of the landing pad or build a better rocket.

We can also distinguish between the modes of failure for the rockets. Namely, Figure 3(c) shows a PCA projection of failures (with $\gamma_{\text{test}} = -15$) onto 2 dimensions. Analysis of the PCA modes

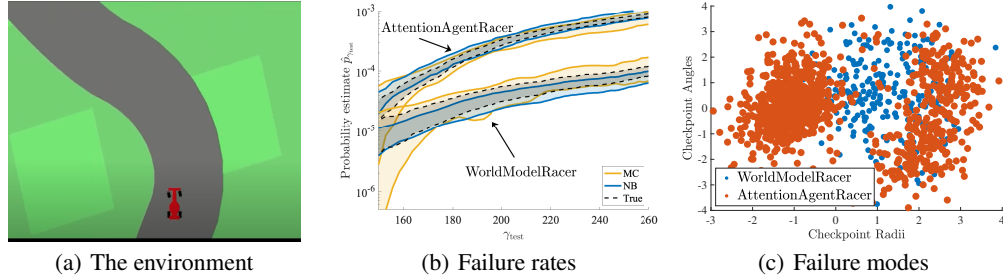

|                  | (a) The environment | (b) Failure rates | (c) Failure modes |

**Figure 4.** CarRacing experiments. MC cannot distinguish between the policies below $\gamma_{\text{test}} = 160$. NB's high-confidence estimates enable model comparisons at extreme limits of failure. Low-dimensional visualization of the failure modes shows that the algorithms fail in distinct ways.

**Table 1:** Relative mean-square error $\mathbb{E}[(\hat{p}_\gamma/p_\gamma - 1)^2]$ over 10 trials

|            | Synthetic          | MountainCar        | Rocket1            | Rocket2            | AttentionAgentRacer       | WorldModelRacer           |
|------------|--------------------|--------------------|--------------------|--------------------|---------------------------|---------------------------|
| MC         | 1.1821             | 0.2410             | 1.1039             | 0.0865             | 1.0866                    | 0.9508                    |
| AMS        | 0.0162             | 0.5424             | 0.0325             | 0.0151             | 1.0211                    | 0.8177                    |
| B          | 0.0514             | 0.3856             | 0.0129             | 0.0323             | 0.9030                    | 0.7837                    |
| NB         | **0.0051**         | **0.0945**         | **0.0102**         | **0.0078**         | **0.2285**                | **0.1218**                |
| $p_\gamma$ | $3.6 \cdot 10^{-6}$ | $1.6 \cdot 10^{-5}$ | $2.3 \cdot 10^{-5}$ | $2.4 \cdot 10^{-4}$ | $\approx 2.5 \cdot 10^{-5}$ | $\approx 9.5 \cdot 10^{-6}$ |

indicates that failures are dominated by high altitude and medium altitude gusts. Even though Rocket2 has a higher probability of failure, its failure mode is more concentrated than Rocket1's failures.

**Car racing**  The CarRacing environment (Figure 4(a)) is a challenging reinforcement-learning task with a continuous action space and pixel observations. Similar observation spaces have been proposed for real autonomous vehicles (*e.g.* [7, 60, 103]). We compare two recent approaches, AttentionAgentRacer [98] and WorldModelRacer [43] that have similar average performance: they achieve average rewards of $903 \pm 49$ and $899 \pm 46$ respectively (mean $\pm$ standard deviation over 2 million trials). Both systems utilize one or more deep neural networks to plan in image-space, so neither has performance guarantees. We evaluate the probability of getting small rewards ($\gamma = 150$).

The 24-dimensional search space $P_0$ parametrizes the generation of the racing track (details are in Appendix C). This environment does not easily provide gradients due to presence of a rendering engine in the simulation loop. Instead, we fit a Gaussian process surrogate model to compute $\nabla f(x)$ (see Appendix C). As these experiments are extremely expensive (taking up to 1 minute per simulation), we only use 2 million naive Monte Carlo samples to compute the ground-truth failure rates. Figure 4(b) shows that, even though the two models have very similar average performance, their catastrophic failure curves are distinct. Furthermore, MC is unable to distinguish between the policies below rewards of 160 due to its high uncertainty, whereas NB clearly shows that WorldModelRacer is superior. Note that, because even the ground-truth has non-negligible uncertainty with 2 million samples, we only report the variance component of relative mean-square error in Table 1.

As with the rocket design experiments, we visualize the modes of failure (defined by $\gamma_{\text{test}} = 225$) via PCA in Figure 4(c). The dominant eigenvectors involve large differentials between radii and angles of consecutive checkpoints that are used to generate the racing tracks. AttentionAgentRacer has two distinct modes of failure, whereas WorldModelRacer has a single mode.

## 5   Conclusion

There is a growing need for rigorous evaluation of safety-critical systems which contain components without formal guarantees (*e.g.* deep neural networks). Scalably evaluating the safety of such systems in the presence of rare, catastrophic events is a necessary component in enabling the development of trustworthy high-performance systems. Our proposed method, neural bridge sampling, employs three concepts—exploration, exploitation, and optimization—in order to evaluate system safety with provable statistical and computational efficiency. We demonstrate the performance of our method on a variety of reinforcement-learning and robotic systems, highlighting its use as a tool for continuous integration and rapid engineering design. In future work, we intend to investigate how efficiently sampling rare failures—like we propose here for *evaluation*—could also enable the *automated repair* of safety-critical reinforcement-learning agents.

## Broader Impact

This paper presents both foundational theory and methods for efficiently evaluating the performance of safety-critical autonomous systems. By definition, such systems can cause injury or death if they malfunction [15]. Thus, improving the tools that practitioners have to perform risk-estimation has the potential to provide a strong positive impact. On the other hand, the improved scalability of our method could be used to more efficiently find (zero-day) exploits and failure modes in $P_0$ (the model of the operational design domain). However, we note that adversarial examples or exploits can also be found via a variety of purely optimization-based methods [3]. The nuances of our method are primarily concerned with the frequency of adverse events, an extra burden; thus, we anticipate they will be of little interest to malicious actors who can manipulate the observations and sensor measurements of complex systems. Another potential concern about the use of our method is with respect to the identification of $P_0$, which we specifically assume to be known in this paper. The gap between $P_0$ in simulation and the real distribution of the environment could lead to overconfidence in the capabilities of the system under test. In Section 1.1 we outline complementary work in anomaly detection and distributionally robust optimization which could mitigate such risks. Still, more work needs to be done to standardize the operational domain of specific tasks by regulators and technology-stakeholders. Nevertheless, we believe that our method will enable the comparison of autonomous systems in a common language—risk—across the spectrum from engineers to regulators and the public.

The applications of our technology are diverse (*cf.* Corso et al. [27]), ranging from testing autonomous vehicles [76, 74] and medical devices [77] to evaluating deep neural networks [104] and reinforcement-learning agents [101]. In the case of autonomous vehicles, Sparrow and Howard [97] argue that it will be morally wrong not to deploy self-driving technology once performance exceeds human capabilities. Our work is an important tool for determining when this performance threshold is achieved due to the rare nature of serious accidents [51]. While the widespread availability of autonomy-enabled devices could narrowly benefit public health, there are many external risks associated with their development. First, many learning-based components of these systems will require massive and potentially invasive data collection [85]; preserving privacy of the public via federated learning [64] and differential privacy-based mechanisms [38] should remain important initiatives within the machine-learning community. A second potential negative consequence of the applications like autonomous vehicles is the use of the real-world as a "simulator" within a reinforcement-learning scheme by releasing "beta" autonomy features (*e.g.* Tesla Autopilot [52]). Unlike established industries such as aerospace [100], many potential applications currently lack regulation and standards; it is important to ensure that industry works with policy makers to develop safety standards in a way that avoids regulatory capture. If widely adopted in regulatory frameworks, our tool would enable rational decisions about the impact, positive or negative, of safety-critical autonomous systems before real lives are affected.

More broadly, the advent of autonomy could spark significant societal changes. For example, the autonomous applications described previously could become core components of weapons systems and military technology that are incompatible with (modern interpretations of) just war theory [96]. Similarly, the automation of the transportation industry has the potential to rapidly destroy the economics of public infrastructure and cost millions of jobs [97]. Thus, Benkler [9] highlights that there is a growing need for the academic community to take action on defining the broader performance criteria to which we will hold AI applications. Brundage et al. [18] and Wing [106] outline broad research agendas which are necessarily interdisciplinary. Still, much more work needs to be done to empower researchers to influence policy. These efforts will require systemic initiatives by research institutions and organizations to engage with local, national, and international governing bodies.

## Acknowledgements

AS and JD were partially supported by the DAWN Consortium, NSF CAREER CCF-1553086, NSF HDR 1934578 (Stanford Data Science Collaboratory), ONR YIP N00014-19-2288, and the Sloan Foundation. MOK was supported by an NSF GRFP Fellowship. RT was supported by Lincoln Laboratory/Air Force Award No. PO# 7000470769 and Amazon Robotics Award No. CC MISC 00272683 2020 TR.

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
