[Supplementary Material]

---

**Algorithm 2** WarpedHMC

---

**Input:** Sample $x$, momentum $v \sim \mathcal{N}(0, I)$, transform $V_\theta$ and its inverse $W_\theta$, scale factor $\beta$, step size $\epsilon$
$y \leftarrow W_\theta(x)$
$v \leftarrow v - 0.5\epsilon\beta I\{f(x) > \gamma\}J_{V_\theta}(y)\nabla f(x)$
$\hat{y} \leftarrow y\cos(\epsilon) + v\sin(\epsilon)$
$\hat{v} \leftarrow v\cos(\epsilon) - y\sin(\epsilon)$
$\hat{x} \leftarrow V_\theta(\hat{y})$
$\hat{v} \leftarrow \hat{v} - 0.5\epsilon\beta I\{f(\hat{x}) > \gamma\}J_{V_\theta}(\hat{y})\nabla f(\hat{x})$
$v \leftarrow -\hat{v}$
$x \leftarrow \hat{x}$ with probability $\min(1, \exp(-H(\hat{x}, \hat{v}) + H(x, v)))$
**Return** $x$

---

## A   Warped Hamiltonian Monte Carlo (HMC)

In this section, we provide a brief overview of HMC as well as the specific rendition, split HMC
[92]. Given "position" variables $x$ and "momentum" variables $v$, we define the Hamiltonian for
a dynamical system as $H(x, v)$ which can usually be written as $U(x) + K(v)$, where $U(x)$ is the
potential energy and $K(v)$ is the kinetic energy. For MCMC applications, $U(x) = -\log(\rho_0(x))$ and
we take $v \sim \mathcal{N}(0, I)$ so that $K(v) = \|v\|^2/2$. In HMC, we start at state $x_i$ and sample $v_i \sim \mathcal{N}(0, I)$.
We then simulate the Hamiltonian, which is given by the partial differential equations:

$$\dot{x} = \frac{\partial H}{\partial v}, \quad \dot{v} = -\frac{\partial H}{\partial x}.$$

Of course, this must be done in discrete time for most Hamiltonians that are not perfectly integrable.
One notable exception is when $x$ is Gaussian, in which case the dynamical system corresponds to the
evolution of a simple harmonic oscillator (*i.e.* a spring-mass system). When done in discrete time, a
symplectic integrator must be used to ensure high accuracy. After performing some discrete steps of
the system (resulting in the state $(x_f, v_f)$), we negate the resulting momentum (to make the resulting
proposal reversible), and then accept the state $(x_f, -v_f)$ using the standard Metropolis-Hastings
criterion: $\min(1, \exp(-H(x_f, -v_f) + H(x_i, v_i)))$ [45].

The standard symplectic integrator—the leap-frog integrator—can be derived using the following
symmetric decomposition of the Hamiltonian (performing a symmetric decomposition retains the
reversibility of the dynamics): $H(x, v) = U(x)/2 + K(v) + U(x)/2$. Using simple Euler integration
for each term individually results in the following leap-frog step of step-size $\epsilon$:

$$v_{1/2} = v_i - \frac{\epsilon}{2}\frac{\partial U(x_i)}{\partial x}$$
$$x_f = x_i + \epsilon\frac{\partial K(v_{1/2})}{\partial v}$$
$$v_f = v_{1/2} - \frac{\epsilon}{2}\frac{\partial U(x_f)}{\partial x},$$

where each step simply simulates the individual Hamiltonian $H_1(x, v) = U(x)/2$, $H_2(x, v) = K(v)$,
or $H_3(x, v) = U(x)/2$ in sequence. As presented by Shahbaba et al. [92], this same decomposition
can be done in the presence of more complicated Hamiltonians. In particular, consider the Hamiltonian
$H(x, v) = U_1(x) + U_0(x) + K(v)$. We can decompose this in the following manner: $H_1(x, v) =$
$U_1(x)/2$, $H_2(x, v) = U_0(x) + K(v)$, and $H_3(x, v) = U_1(x)/2$. We can apply Euler integration to
the momentum $v$ for the first and third Hamiltonians and the standard leap-frog step to the second
Hamiltonian (or even analytic integration if possible). For this paper, we have $U_0(x) = -\log\rho_0(x)$
and $U_1(x) = -\beta[\gamma - f(x)]_-$.

To account for warping, the modifications needed to the HMC steps above are simple. When
performing warping, we simply perform HMC for a Hamiltonian $\hat{H}(y, v)$ that is defined with respect
to the warped position variable $y$, where $x = V_\theta(y)$ for given parameters $\theta$. By construction of
the normalizing flows, we assume $y \sim \mathcal{N}(0, I)$, so that we can perform the dynamics for $\hat{H}_2(y, v)$
analytically. Furthermore, the Jacobian $J_{V_\theta}(y)$ is necessary for performing the Euler integration
of $\hat{H}_1(y, v)$ and $\hat{H}_3(y, v)$. This is summarized in Algorithm 2. Note that we always perform the
Metropolis-Hastings acceptance with respect to the true Hamiltonian $H$, rather than the Hamiltonian
$\hat{H}$ that assumes perfect training of the normalizing flows.

**HMC and non-smooth functions** In Section 2, we assumed that the measure of non-differentiable points is zero for the energy potentials considered by HMC. As discussed by Afshar and Domke [2], the inclusion of the Metropolis-Hastings acceptance criterion as well as the above assumption ensures that HMC asymptotically samples from the correct distribution even for non-smooth potentials. An equivalent intuitive explanation for this can be seen by viewing the ReLU function $[x]_+$ as the limit of softplus functions $g_k(x) := \log(1 + \exp(kx))/k$ as the sharpness parameter $k \to \infty$. We can freely choose $k$ such that, up to numerical precision, Algorithm 2 is the same whether we consider using a ReLU or sufficiently sharp (*e.g.* large $k$) softplus potential, because, with probability one, we will not encounter the points where the potentials differ. When further knowledge about the structure of the non-differentiability is known, the acceptance rate of HMC proposals can be improved [78, 57, 79, 2, 21].

# B  Performance analysis

## B.1  Proof of Proposition 1

We begin with showing the convergence of the number of iterations. To do this, we first show almost sure convergence of $\beta_k$ in the limit $N \to \infty$. We note that in the optimization problem (8), $\beta_k$ is a feasible point, yielding $b_k(\beta) = 1$. Thus, $\beta_{k+1} \geq \beta_k \geq \beta_0 := 0$. Due to this growth of $\beta_k$ with $k$, we have

$$\frac{Z_{k+1}}{Z_k} = \mathbb{E}_{P_k}\left[\frac{\rho_{k+1}(X)}{\rho_k(X)}\right] \leq 1,$$

$$\mathbb{P}_k(f(X) \leq \gamma) = \mathbb{E}_{P_{k+1}}\left[\frac{Z_{k+1}}{Z_k}\frac{\rho_k(X)}{\rho_{k+1}(X)}I\{f(X) \leq \gamma\}\right]$$

$$= \frac{Z_{k+1}}{Z_k}\mathbb{E}_{P_{k+1}}\left[I\{f(X) \leq \gamma\}\right]$$

$$\leq \mathbb{P}_{k+1}(f(X) \leq \gamma).$$

By the uniform convergence of empirical measures offered by the Glivenko-Cantelli Theorem, the value $a_k \to \mathbb{P}_k(f(X) \leq \gamma)$ almost surely. Then, the stop condition can be rewritten as $b_k(\beta) \geq a_k/s \to \mathbb{P}_k(f(X) \leq \gamma)/s \geq p_\gamma/s$. Since $b_k(\beta)$ is monotonically decreasing in the quantity $\beta - \beta_k$, this constraint gives an upper bound for $\beta_{k+1}$, and, as a result, all $\beta_k$ are almost surely bounded from above and below. We denote this interval as $\mathcal{B}$.

Now, we consider the convergence of the solutions to the finite $N$ versions of problem (8), denoted $\beta_k^N$, to the "true" optimizers $\beta_k$ in the limit as $N \to \infty$. Leaving the dependence on $\beta_k$ implicit for the moment, we consider the random variable $Y := g(X; \beta) := \exp\left((\beta - \beta_k)[\gamma - f(X)]_-\right)$. Then, since $\beta \in \mathcal{B}$ is bounded and $g$ is continuous in $\beta$, we can state the Glivenko-Cantelli convergence of the empirical measure uniformly over $\mathcal{B}$: $\sup_{\beta \in \mathcal{B}} \|F^N(Y) - F(Y)\|_\infty \to 0$ almost surely, where $F$ is the cumulative distribution function for $Y$. Note that the constraints in the problem (8) can be rewritten as expectations of this random variable $Y$. Furthermore, the function $g$ is strictly monotonic in $\beta$ (and therefore invertible) for non-degenerate $f(X)$ (*i.e.* $f(x) > \gamma$ for some non-negligible measure under $P_0$). Thus, we have almost sure convergence of the argmin $\beta_{k+1}^N$ to $\beta_{k+1}$.

Until now, we have taken dependence on $\beta_k$ implicitly. Now we make the dependence explicit to show the final step of convergence. In particular, we can write $\beta_{k+1}$ as a function of $\beta_k$ (along with their empirical counterparts), For concreteness, we consider the following decomposition for two iterations:

$$|\beta_2^N(\beta_1^N) - \beta_2(\beta_1)| \leq |\beta_2^N(\beta_1^N) - \beta_2(\beta_1^N)| + |\beta_2(\beta_1^N) - \beta_2(\beta_1)|.$$

We have already shown above that the first term on the right hand side vanishes almost surely. By the same reasoning, we know that $\beta_1^N \to \beta_1$ almost surely. The second term also vanishes almost surely since $\beta_{k+1}(\beta)$ is a continuous mapping. This is due to the fact that the constraint functions in problem (8) are continuous functions of both $\beta$ and $\beta_k$ along with the invertibility properties discussed previously. Then, we simply extend the telescoping series above for any $k$ and similarly show that all terms vanish almost surely. This shows the almost sure convergence for all $\beta_k$ up to some $K$.

Now we must show that $K$ is bounded and almost surely converges to a constant. To do this we explore the effects of the optimization procedure. Assuming the stop condition (the second constraint)

does not activate, the first constraint in problem (8) has the effect of making $\mathbb{Z}_{k+1}/Z_k = \alpha$ (almost surely), which implies $\mathbb{P}_{k+1}(f(X) \leq \gamma) = \mathbb{P}_k(f(X) \leq \gamma)/\alpha$. In other words, we magnify the event of interest by a factor of $1/\alpha$. The second constraint can be rewritten as $\mathbb{P}_{k+1}(f(X) \leq \gamma) \leq s$. Thus, we magnify the probability of the region of interest by factors of $\alpha$ unless doing so would increase the probability to greater than $s$. In that case, we conclude with setting the probability to $s$ (since $\mathbb{P}_\beta(f(X) \leq \gamma)$ is monotonically increasing in $\beta$). In this way, we have 0 iterations for $p_\gamma \in [s, 1]$, 1 iteration for $p_\gamma \in [\alpha s, s)$, 2 iterations for $p_\gamma \in [\alpha^2 s, \alpha s)$, and so on. Then, the total number of iterations is (almost surely) $\lfloor \log(p_\gamma)/\log(\alpha) \rfloor + I\{p_\gamma/\alpha^{\lfloor \log(p_\gamma)/\log(\alpha) \rfloor} < s\}$.

Now we move to the relative mean-square error of $\hat{p}_\gamma$. We employ the delta method, whereby, for large $N$, this is equivalent to $\mathrm{Var}(\log(\hat{p}_\gamma))$ (up to terms $o(1/N)$). For notational convenience, we decompose $\widehat{E}_k$ into its numerator and denominator:

$$A_k(X) := \rho_k^B(X)/\rho_{k-1}(X), \qquad \widehat{A}_k := \frac{1}{N}\sum_{i=1}^N A_k(x_i^{k-1})$$

$$B_k(X) := \rho_k^B(X)/\rho_k(X), \qquad \widehat{B}_k := \frac{1}{N}\sum_{i=1}^N B_k(x_i^k).$$

By construction (and assumption of large $T$), Algorithm 1 has a Markov property that each iteration's samples $x_i^k$ are independent of the the previous iterations' samples $x_i^{k-1}$ given $\beta_k$. For shorthand, let $\beta_{0:k}$ denote all $\beta_0, \ldots, \beta_k$. Conditioning on $\beta_{0:k}$, we have

$$\mathrm{Var}(A_k) = \mathrm{Var}\left(\mathbb{E}[A_k|\beta_{0:k}]\right) + \mathbb{E}\left[\mathrm{Var}\left(A_k|\beta_{0:k}\right)\right].$$

Since $\beta_{0:k}$ approaches constants almost surely as $N \to \infty$, the first term vanishes and the second term is the expectation of a constant. In particular, the second term is as follows:

$$\mathrm{Var}\left(A_k|\beta_{0:k}\right) = \mathbb{E}\left[A_k^2|\beta_{0:k}\right] - \left(\mathbb{E}\left[A_k|\beta_{0:k}\right]\right)^2$$

$$= \mathbb{E}_{P_{k-1}}\left[\frac{\rho_k(X)}{\rho_{k-1}(X)}\right] - \left(\mathbb{E}_{P_{k-1}}\left[\sqrt{\frac{\rho_k(X)}{\rho_{k-1}(X)}}\right]\right)^2$$

$$= \frac{Z_k}{Z_{k-1}} - \left(\frac{Z_k^B}{Z_{k-1}}\right)^2.$$

Similarly, $\mathrm{Var}(B_k|\beta_{0:k}) = Z_{k-1}/Z_k - (Z_k^B/Z_k)^2$. Next we look at the covariance terms:

$$\mathrm{Cov}(A_{k-1}, A_k) = \mathrm{Cov}\left(\mathbb{E}[A_{k-1}|\beta_{0:k}], \mathbb{E}[A_k|\beta_{0:k}]\right) + \mathbb{E}\left[\mathrm{Cov}\left(A_{k-1}, A_k|\beta_{0:k}\right)\right].$$

Again, the first term vanishes since $\beta_{0:k}$ approach constants as $N \to \infty$. By construction, the second term is also 0 since the quantities are conditionally independent. Similarly, $\mathrm{Cov}(B_{k-1}, B_k) = 0$ and $\mathrm{Cov}(A_i, B_j) = 0$ for $j \neq i-1$. However, there is a nonzero covariance for the quantities that depend on the same distribution:

$$\mathrm{Cov}\left(B_k, A_{k+1}|\beta_{0:k+1}\right) = \mathbb{E}\left[B_k A_{k+1}|\beta_{0:k+1}\right] - \mathbb{E}\left[B_k|\beta_{0:k+1}\right]\mathbb{E}\left[A_{k+1}|\beta_{0:k+1}\right]$$

$$= \mathbb{E}_{P_k}\left[\frac{\sqrt{\rho_{k-1}(X)\rho_{k+1}(X)}}{\rho_k(X)}\right] - \frac{Z_{k+1}^B}{Z_k}\frac{Z_k^B}{Z_k}$$

$$= \frac{Z_k^C}{Z_k} - \frac{Z_{k+1}^B}{Z_k}\frac{Z_k^B}{Z_k}.$$

By the large $T$ assumption, the samples $x_i^k$ and $x_j^k$ are independent for all $i \neq j$ given $\beta_k$. Then we have

$$\mathrm{Var}(\widehat{A}_k|\beta_{0:k}) = \mathrm{Var}(A_k|\beta_{0:k})/N, \quad \mathrm{Var}(\widehat{B}_k|\beta_{0:k}) = \mathrm{Var}(B_k|\beta_{0:k})/N,$$

$$\mathrm{Cov}(\widehat{B}_k, \widehat{A}_{k+1}|\beta_{0:k+1}) = \mathrm{Cov}(B_k, A_{k+1}|\beta_{0:k+1})/N.$$

The last term in $\hat{p}_\gamma$, $\frac{1}{N}\sum_{i=1}^N \frac{\rho_\infty(x_i^K)}{\rho_K(x_i^K)}$, reduces to a simple Monte Carlo estimate since $\frac{\rho_\infty(X)}{\rho_K(X)} = I\{f(X) \leq \gamma\}$. Furthermore, this quantity is independent of all other quantities given $\beta_{0:K}$ and, as noted above, approaches $s$ almost surely as $N \to \infty$.

Putting this all together, the delta method gives (as $N \to \infty$ so that $\beta_{0:K}$ approach constants almost surely),

$$\text{Var}(\log(\hat{p}_\gamma)) \to \sum_{k=1}^{K} \left( \frac{\text{Var}(\widehat{A}_k)}{(Z_k^B/Z_{k-1})^2} + \frac{\text{Var}(\widehat{B}_k)}{(Z_k^B/Z_k)^2} \right) - 2 \sum_{k=1}^{K-1} \frac{\text{Cov}(\widehat{B}_k, \widehat{A}_{k+1})}{Z_{k+1}^B Z_k^B / Z_k^2} + \frac{1-s}{sN} + o\left(\frac{1}{N}\right).$$

The Bhattacharrya coefficient can be written as

$$G(P_{k-1}, P_k) = \int_{\mathcal{X}} \sqrt{\frac{\rho_{k-1}(x)}{Z_{k-1}} \frac{\rho_k(x)}{Z_k}} \, dx = \frac{Z_k^B}{\sqrt{Z_{k-1} Z_k}}.$$

Furthermore, we have

$$\frac{G(P_{k-1}, P_{k+1})}{G(P_{k-1}, P_k) G(P_k, P_{k+1})} = \frac{Z_k^C}{\sqrt{Z_{k-1} Z_{k+1}}} \frac{\sqrt{Z_{k-1} Z_k}}{Z_k^B} \frac{\sqrt{Z_k Z_{k+1}}}{Z_{k+1}^B} = \frac{Z_k^C Z_k}{Z_k^B Z_{k+1}^B},$$

yielding this final result

$$\text{Var}(\log(\hat{p}_\gamma)) \to \frac{2}{N} \sum_{k=1}^{K} \left( \frac{1}{G(P_{k-1}, P_k)^2} - 1 \right) - \frac{2}{N} \sum_{k=1}^{K-1} \left( \frac{G(P_{k-1}, P_{k+1})}{G(P_{k-1}, P_k) G(P_k, P_{k+1})} - 1 \right) + \frac{1-s}{sN} + o\left(\frac{1}{N}\right). \quad (12)$$

We remark that a special case of this formula is for $K = 1$ and $s = 1$ (so only the first term survives), which is the relative mean-square error for a single bridge-sampling estimate $\widehat{E}_k$.

Now, since $G(P, Q) \geq 0$, the terms in the second sum are $\geq -1$ so that the second sum is $\leq 2(K - 1)/N$. Furthermore, since $s \geq 1/3$, the last term is also $\leq 2/N$. Thus, if we have $\frac{1}{G(P_{k-1}, P_k)^2} \leq D$ (with $D \geq 1$), then the asymptotic relative mean-square error (12) is $\leq 2KD/N$ (up to terms $o\left(\frac{1}{N}\right)$).

When performing warping, we follow the exact same pattern as the above results, conditioning on both $\beta_{0:k}$ and $W_{0:k}$, where $W_0$ is defined as the identity mapping. We follow the same almost-sure convergence proof for $W_k$ as above for $\beta_k$, which requires compactness of $\theta \in \Theta$, continuity of $W$ with respect to $\theta$ and $x$, and that we actually achieve the minimum in problem (6). Although the first two conditions are immediate in most applications, the last condition can be difficult to satisfy for deep neural networks due to the nonconvexity of the optimization problem.

## C  Experimental setups

### C.1  Hyperparameters

The number of samples $N$ affects the absolute performance of all of the methods tested, but not their relative performance with respect to each other. For all experiments, we use $N = 1000$ for B and NB to have adequate absolute performance given our computational budget (see below for the computing architecture used). Other hyperparameters were tuned on the synthetic problem and fixed for the rest of the experiments (with the exception of the MAF architecture for the rocket experiments). The hyperparameters were chosen as follows.

When performing Hamiltonian dynamics for a Gaussian variable, a time step of $2\pi$ results in no motion and time step of $\pi$ results in a mode reversal, where both the velocity and position are negated. The $\pi$ time step is in this sense the farthest exploration that can occur in phase space (which can be intuitively understood by recognizing that the phase diagram of a simple spring-mass system is a unit circle). Thus, we considered $T = 4, 8, 12$, and $16$ with time steps $\pi/T$. We found that $T = 8$ provided reasonable exploration (as measured by autocorrelations and by the bias of the final estimator $\hat{p}_\gamma$) and higher values of $T$ did not provide much more benefit. For B, we allowed 2 more steps $T = 10$ to keep the computational cost the same across B and NB. Similarly, for AMS, we set $T = 10$. We also performed tuning online for the time step to keep the acceptance ratio between 0.4 and 0.8. This was done by setting the time step to $\sin^{-1}(\min(1, \sin(t) \exp((p - C)/2))$, where $t$ is the current time step, $p$ is the running acceptance probability for a single chain and $C = 0.4$ if $p < 0.4$ or 0.8 if $p > 0.8$. This was done after every $T$ HMC steps.

For the step size of the bridge, we considered $\alpha \in \{0.01, 0.1, 0.3, 0.5\}$. Smaller $\alpha$ results in fewer iterations and better computational efficiency. However, we found that very small $\alpha$ made MAF

training difficult (see below for the MAF architectures used). We settled on $\alpha = 0.3$, which provided reasonable computational efficiency (no more than 11 iterations for the synthetic problem) as well as stable MAF training. For AMS, we followed the hyperparameter settings of Webb et al. [104]. Namely, we chose a culling fraction of $\alpha_{\text{AMS}} = 10\%$, where $\alpha_{\text{AMS}}$ sets the fraction of particles that are removed and rejuvenated at each iteration [104].

The MAF architectures for the synthetic, MountainCar, and CarRacing experiments were set at 5 MADE units, each with 1 hidden layer of 100 neurons. Because the rocket search space is very high dimensional, we decreased the MAF size for computational efficiency: we set it at 2 MADE units, each with hidden size 400 units. We used 100 epochs for training, a batch size of 100, a learning rate of 0.01 and an exponential learning-rate decay with parameter 0.95.

Given the above parameters, the number of simulations for each experiment varies based on the final probability in question $p_\gamma$ (smaller values result in more simulations due to having a higher number of iterations $K$). We had runs of 111000, 101000, 91000, 71000, 91000, and 101000 simulations respectively for the synthetic, MountainCar, Rocket1, Rocket2, AttentionAgentRacer, and WorldModelRacer environments. We used these values as well as the ground truth $p_\gamma$ values to determine the number of particles allowed for AMS, $N_{\text{AMS}} = 920, 910, 820, 780, 820, 910$ respectively, as AMS has a total cost of $N_{\text{AMS}}(1 + \alpha_{\text{AMS}}TK_{\text{AMS}})$, where $K_{\text{AMS}} \approx \log(p_\gamma)/\log(1 - \alpha_{\text{AMS}})$.

For the surrogate Gaussian process regression model for CarRacing, we retrained the model on the most recent $N$ simulations after every $NT$ simulations (*e.g.* after every $T$ HMC iterations). This made the amortized cost of training the surrogate model negligible compared to performing the simulations themselves. We used a Matern kernel with parameter $\nu = 2.5$. We optimized the kernel hyperparameters using an L-BFGS quasi-Newton solver.

**Computing infrastructure and parallel computation** Experiments were carried out on commodity CPU cloud instances, each with 96 Intel Xeon cores @ 2.00 GHz and 85 GB of RAM. AMS, B, and NB are all designed to work in a Map-Reduce paradigm, where a central server orchestrates many worker jobs followed by synchronization step. AMS requires more iterations and fewer parallel worker threads per iteration than B and NB. In particular, whereas B and NB perform $N$ parallel jobs per iteration, AMS only performs $\alpha_{\text{AMS}}N_{\text{AMS}}$ parallel jobs per iteration. Thus, B and NB take advantage of massive scale and parallelism much more than AMS.

## C.2 Environment details

### C.2.1 MountainCar

The MountainCar environment considers a simple car driving on a mountain road. The car can sense horizontal distance $s$ as well as its velocity $v$, and may send control inputs $u$ (the amount of power applied in either the forward or backward direction). The height of the road is given by: $h(s) = 0.45 \sin(3s) + 0.55$. The speed of the car, $v$, is a function of $s$ and $u$ only. Thus, the discrete time dynamics are: $s_{k+1} = s_k + v_{k+1}$ and $v_{k+1} = v_k + 0.0015u_k - 0.0025\cos(3s_k)$. For a given episode the agent operating the car receives a reward of $-0.1u_k^2$ for each control input and 100 for reaching the goal state.

In this experiment we explore the effect of domain shift on a formally verified neural network. We utilize the neural network designed by Ivanov et al. [48]; it contains two hidden layers, each of 16 neurons, for a total of 337 parameters. For our experiments we use the trained network parameters available at: `https://github.com/Verisig/verisig`. Ivanov et al. [48] describe a layer-by-layer approach to verification which over-approximates the reachable set of the combined dynamics of the environment and the neural network. An encoding of this system (network and environment) is developed for the tool Flow* [24] which constructs the (overapproximate) reachable set via a Taylor approximation of the combined dynamics.

The MountainCar environment is considered solved if a policy achieves an average reward of 90 over 100 trials. The authors instead seek to prove that the policy will achieve a reward of at least 90 for any initial condition. By overapproximating the reachable states of the system, they show that the car always receives a total reward greater than 90 and achieves the goal in less than 115 steps for a subset of the intial conditions $\hat{p}_0 \in [-0.59, -0.4]$.

### C.2.2 Rocket design

The system under test is a rocket spacecraft with dynamics $m\ddot{p} = f - mge_3$ , where $m > 0$ is the mass, $p(t) \in \mathbf{R}^3$ is the position, and $e_3$ is the unit vector in the z-direction. While it is possible to synthesize optimal trajectories for an idealized model of the system, significant factors such as wind and engine performance (best modeled as random variables) are unaccounted for [13]. Without feedback control, even small uncorrected tracking errors result in loss of the vehicle. In the case of disturbances the authors suggest two approaches: (1) a feedback control law which tracks the optimal trajectory (2) receding horizon model predictive control. The system we consider tracks an optimal trajectory using a feedback control law. Namely, the optimal trajectory is given by the minimum fuel solution to a linearized mode of the dynamics. Specifically, we consider the thrust force discretized in time with a zero-order hold, such that $f_k$ applied for time $t \in [(k-1)h, kh]$ for a time step $h = 0.2$. Then, the reference thrust policy solves the following convex optimization problem

$$\text{minimize} \sum_{i=1}^{K} \|f_k\|_2$$
$$\text{such that } p_K = v_K = 0, \|f_k\| \leq F_{\max},$$
$$v_{k+1} - v_k = \frac{h}{m} f_k - hge_3,$$
$$p_{k+1} - p_k = \frac{h}{2}(v_k + v_{k+1}),$$
$$(p_3)_k \geq 0.5\|((p_1)_k, (p_2)_k)\|_2,$$

where the last constraint is a minimum glide slope and $F_{\max}$ is a maximum thrust value for the nominal thrusters. This results in the thrust profile $f^\star$. The booster thrusters correct for disturbances along the flight. The disturbances at every point in time follow a mixture of Gaussians. Namely, we consider 3 wind gust directions, $w_1 = (1,1,1)/\sqrt{(3)}$, $w_2 = (0,1,0)$, and $w_3 = (1,0,0)$. For every second in time, the wind follows a mixture:

$$W \sim \mathcal{N}(0, I) + w_1 B + w_2 \hat{B} + (1 - \hat{B})w_3,$$

where $B \sim \text{Bernoulli}(1/3)$ and $\hat{B} \sim \text{Bernoulli}(1/2)$. This results in 5 random variables for each second, or a total of 100 random variables since we have a 20 second simulation. The wind intensity experienced by the rocket is a linear function of height (implying a simplistic laminar boundary layer): $f_w = CWp_3$ for a constant $C$. Finally, the rocket has a proportional feedback control law for the booster thrusters to the errors in both the position $p_k$ and velocity $v_k$:

$$f_{\text{feedback,k}} = \text{clip-by-norm}(f_k^\star - K_p(p_k - p_k^\star) - K_v(v_k - v_k^\star)).$$

The maximum norm for clip-by-norm is $aF_{max}$, where $a = 1.15$ for Rocket1 and $a = 1.1$ for Rocket2, indicating that the boosters are capable of providing 15% or 10% of the thrust of the main engine.

### C.2.3 Car Racing

We compare the failure rate of agents solving the car-racing task utilizing the two distinct approaches ([43] and [98]). The car racing task differs from the other experiments due to the inclusion of a (simple) renderer in the system dynamics. At each the step the agent recieves a reward of $-0.1 + \mathcal{I}_{newtile}(1000/N) - \mathcal{I}_{offtrack}(100)$ where N is the total number of tiles visited in the track. The environment is considered solved if the agent returns an average reward of 900 over 100 trials. The search space $P_0$ is the inherent randomness involved with generating a track. The track is generated by selecting 12 checkpoints in polar coordinates, each with radian value uniformly in the interval $[2\pi i/12, 2\pi(i+1)/12)$ for $i = 0, \ldots 11$, and with radius uniformly in the interval $[R/3, R]$, for a given constant value $R$. This results in 24 parameters in the search space. The policies used for testing are described below (with training scripts in the code supplement).

**AttentionAgent**   Tang et al. [98] utilize a simple self-attention module to select patches from a 96x96 pixel observation. First the input image is normalized then a sliding window approach is used to extract N patches of size $M \times M \times 3$ which are flattened and arranged into a matrix of size $3M^2 \times N$. The self-attention module is used to compute the attention matrix $A$ and importance vector

(summation of each column of $A$). A feature extraction operation is applied to the top K elements of the sorted importance vector and the selected features are input to a neural network controller. Both the attention module and the controller are trained together via CMA-ES. Together, the two modules contain approximately 4000 learnable parameters. We use the pre-trained model available here: https://github.com/google/brain-tokyo-workshop/tree/master/AttentionAgent.

**WorldModel** The agent of Ha and Schmidhuber [43] first maps a top-down image of the car on track via a variational autoencoder to a latent vector $z$. Given $z$, the world model $M$ utilizes a recurrent-mixture density network [12] to model the distribution of future possible states $P(z_{t+1} \mid a_t, z_t, h_t)$. Note that $h_t$, the hidden state of the RNN. Finally, a simple linear controller $C$ maps the concatenation of $z_t$ and $h_t$ to the action, $a_t$. We use the pre-trained model available here: https://github.com/hardmaru/WorldModelsExperiments/tree/master/carracing.