[Reviews · NeurIPS 2020]

Review 1

Summary and Contributions: Summary of contributions ==================== i) They set out to deploy probabilistic methods to determine the probability of dangerous events and determine the safety of a given, where dangerous events are simulated in a custom-built simulator, that combines exploration, exploitation, and optimization techniques to find failure modes and estimate the rate of occurrence. ii) They provide weak guarantees for the performance of their methods in terms of computational and statistical efficiency, shown via comprehensive numerical experiments. iii) Deploy their techniques on some interesting scenarios and demonstrate the effectiveness of their techniques. Summary ======= They combine an adapted version of HMC, that they call warped HMC which, through sequential updates, utilizes normalizing flows and bridge sampling to extract samples corresponding to rare-events in a variety of different scenarios, generated via stochastic simulation. This paper shares some similar themes with NeuTra-lizing Bad Geometry in Hamiltonian Monte Carlo Using Neural Transport, but they also combine a series of other techniques. ***** UPDATE ***** I have read the rebuttal and thank the authors for their updates. I had read this two-weeks ago and contributed to the discussions, so I apologise for the delay in the update. Just a few points and I believe the AC/ other reviewers have provided you with more feedback. The ReLu in the potential function induces discontinuities and violates the smoothness of the potential and reversibility criteria of HMC. There is a quick a dirty fix for the reversibility problem, which will make your sampling even more inefficient, and that is to put a stopping criteria on the leapfrog step every time your sample becomes negative - you will need to add a check for this in the code. See the STAN manual for ways to make this efficient. To address the smoothness, well, the ReLU function induces a measure zero discontinuity so it is "kind-of-okay" in the loosest sense. Please don't deploy this algorithm on any safety critical procedures. I am giving you the benefit of the doubt here, hence my increase in score, as you have done a lot of work for this application and I do like the application - so please make the fix in the leapfrog step in the code base. Furthermore, you will need to take a lot more MC samples for your estimates to be valid and maybe you have already done so, but I couldn't see any tests in your code base, run your inference schemes on a model that you can alanytically verify that has a density of a similar form, but a much simpler problem. Then you can be sure that things actually work.

Strengths: They explicitly use first-order information about f to improve sampling and accelerate the convergence of the estimate through optimization, as opposed to methods such as multi-level splitting that don’t. They state that second-order information is required. However, their method does not require it, mitigating the need to do, slightly more expensive, second-order gradient estimates. On the one hand, this paper fuses lots of different techniques to try and approach the interesting problem of understanding when a system will fail. They do this by developing a novel inference scheme to extract samples corresponding to those scenarios. This is challenging because they can often be in the tails of distributions, or be defined by complicated geometries, which makes sampling challenging unless you are willing to incorporate the full geometric structure, which requires second-order gradients, which is computationally expensive. The work they are doing is important and definitely of importance to the NeurLPS community; however, see weaknesses.

Weaknesses: What confuses me about this paper is that they state they avoid having to do complex hessian evaluations, yet use normalizing flows, which involve lots of jacobian calculations and matrix inversions. I don't feel as though their justifications in section 3 are strong enough for this claim. Also, I have some concerns surrounding the inference performed for their simulators, see correctness box. Also, I find many parts of the paper incoherent, and I don't fully appreciate the messages. I don't mean incoherent in a terrible way; it is just the message that seems convoluted at times. I think that this paper does contain some novel contributions and is of significance. Still, the way aspects of the paper are written, for example, section 3, make this paper hard to digest and appreciate. They have tried to be comprehensive in covering related work, but have also missed important work related to this problem. None of the specific methods they are using are novel, but combining them is novel.

Correctness: It seems so, but there are one or two-pieces of strange notation, which makes it challenging to be entirely sure what the log density of the Hamiltonian is, and I don't want to second guess anything. I made a note in the clarity box. I should emphasize that I think this is to do with the way the methods have been presented, not the authors' work, which seems, at least from following most of the maths, correct. Of course, I have not been able to run those experiments as the servers I have access to do not allow docker for security concerns. I am also unsure if the simulators contain nested structures, see Nested Probabilistic Programs, Rainforth. If they did, which is common in stochastic simulators like these, then this would change my evaluation significantly as the ground truth is generated via an MC estimate, and 5*10^7, for more complicated problems, is not actually that many samples. There are not enough details in the text (inc supplement), and it is difficult to extract these details from the code.

Clarity: See the comments above. Here are some extra points: Line 17: agent better than a human at .. -> agent, that supersedes human ability at … Line 20: Add PID acronym Line 25: efficiently evaluate complex systems -> as you are focusing on the statistical element and you state in the abstract (computationally and statistically efficient) it may be a good idea to emphasize that point here. Since efficiently is a vague term. Line 31: Get rid of the ref to [33] and replace it with a proper reference from the statistics-based generative modelling community. I do not think that is a good reference here. Line 68: to -> for Line 92: ...measures probabilities of rare events inefficiently.. - I do not like the use of measures here, maybe “calculates”, or “. naive MC is inefficient at exploring the tails of distributions, making it challenging to determine the probabilities of rare-events … Line 93: Sequential monte carlo, this is mis-leading as you are not doing SMC, at least, it does not look like that - maybe I am incorrect. See An introduction to sequential Monte Carlo methods ,A Doucet, N De Freitas, N Gordon Equation (3) can you explain that the minus sign on the outer square bracket is? It seems to become mid-bracket in line 556 and low-bracket in other parts. Line 116-117: These dynamics conserve volume in the augmented state-space, even when performed with discrete-time steps Line 121-122: Most importantly, we avoid any need for Hessian computation because the dynamics conserve volume. -- This sentence needs more information. Algorithm 1 - Tidy it up. By which I mean, add <func name> (args) … initialization … then while ... ... I found all of section 3 difficult to get through and quite confusing, I just want to know what the computational cost is and the statistical costs and any assumptions made. The rest can go in the appendix.

Relation to Prior Work: 8/10 I think that the authors do an excellent job of talking about related/ prior work and how their work differs. Based on their assessment of previous work, I have a good understanding of how they improve upon earlier work. But, there are a number of works missing, see additional feedback. However, I do not believe this was intentional.

Reproducibility: Yes

Additional Feedback: Line 35: You cite 18, but a lot of work has been done on sampling from long-tailed distributions, for example, Umbrella sampling: a powerful method to sample tails of distributions , Charles Matthews et al. ; MARKOV CHAIN MONTE CARLO FOR COMPUTING RARE-EVENT PROBABILITIES FOR A HEAVY-TAILED RANDOM WALK, THORBJÖRN GUDMUNDSSON and HENRIK HULT (sorry for the caps), Journal of Applied Probability, Vol. 51, No. 2 (JUNE 2014), pp. 359-376 ; Multicanonical MCMC for sampling rare events: an illustrative review, Yukito Iba, Nen Saito & Akimasa Kitajima; Rare-Event Simulation James L. Beck and Konstantin M. Zuev. If you provide me with some additional pointers and more clarification about the general work, then I will increase my score.


Review 2

Summary and Contributions: I appreciate the addressing of my concerns about organization, prior work, and relationship of theory to experiments. I like this work and congratulate the authors on it. 'On the title: you could just go with "Neural Bridge Sampling". Or "Neural bridge sampling and Evaluation of safety-critical autonomous systems" R1 raised concerns about ReLUs in the flow producing invalid samples. My thought about this is that this would introduce variance but not bias (although I'm not positive about this, it would depend on the init), and if you were using early stopping, the distance from valid samples would be bounded enough not to be a problem in practice. R1 pointed out you're not using ES, and elaborated that the concerns are not just about variance - I do not entirely understand the particulars of the issue, but I understand the overall concern that the ML community could (maybe is) suffer from a 'castle-built-on-sand' problem where we put together a bunch of things that are basically working fine but not quite for the reasons we expect, and it's important to address this type of thing before it "gets too far". I wish there was another round of reviewing so that I could hear the author response to this issue. Overall, I have decided to maintain my score of 6, and leave it to the AC to assess (I also have the impression from the author response that you would care about this type of issue and making sure it is addressed well, so I'm also leaving it to some good-faith on your part). ===================== The work proposes an iterative MCMC-based method for sampling/estimating/predicting the occurrence of "failure modes". The novelty in NBS is that the iterative transformations in the chain are first importaince-sampled and then corrected by a neural density model, which estimates ratios of normalizing constants without needing to compute the Hessian. The authors apply this technique to evaluate the safety of various simple RL systems - essentially defining "safe" as "within the bounds of normal behaviour".

Strengths: I think somewhat "blackbox" testing of AI systems in this way is extremely important. Although it is not put this way explicitly (and I think it could/maybe should be), defining safe behaviour as being within the bounds of testbed performance, where "being within the bounds" is evaluated in a soft/smart way (in this case, the neural density estimates) is a powerful paradigm for this kind of testing, which is not commonplace knowledge in the ML community. The connections made to other safety-testing fields (e.g. formal verification) are valuable and significant in a similar way. The mix of theory/explanation and experiments is just right, and the experiments are adequate to demonstrate points of the proposed approach.

Weaknesses: - some of the jargon/presentation makes this work not very accessible to a neurips audience / the people who would most benefit from really understanding the importance of this type of work and how to do this / apply it in practice. See comments on "clarity". - related work section misses what I think are significantly related fields - organization of the paper feels like it takes a long time to get to the experiments; maybe "performance analysis" could become a subsection of "proposed approach", and/or could be renamed "Theoretical analysis". It could also help if the theory and experiments were more explicitly related to each other. - almost no reproducibility details are given for the experiments, and no mention of code being available

Correctness: - In the abstract the claim is the technique finds failure modes and estimates the rate of their occurance, but as far as I can tell they only do the latter (adverse events and their thresholds are predefined, it is only the rate of occurance that is estimated. The failures aren't clustered or anything to "find failure modes"; maybe this isn't what is meant but I find it misleading -

Clarity: The paper has a lot of jargon and offhand references to fields and concepts that would be unfamiliar to most of the neurips audience. The explanations also seem oriented to that audience, making them somewhat opaque to me sometimes. Some examples: - "exploiting the most informative portions of random samples" is not clear (also not clear how it differs from optimization) - p 44-52 is good to explain the details of why we would want 2nd order information, but it's not clear why volume distortion depending on the hessian has anything to do with the previous paragraph; changing the first sentence to have less jargon (e.g. just "These optimization methods often require second-order information to work well" and then go into the more technical stuff). - "expontial tilting barrier" introduced before explained, "barrier" is mentioned several times without definition - "closed-loop performance" not defined - "definition of the evaluation problem" not familiar wording for an ML audience - surrogate model and most of the "overall efficiency" section The remarks/intuition about the theory are nice and seem correct to me, but they are not linked back to the empirical stuff/intuition about the problem - how tight is the bound? How does it compare to what is found empirically? "Quantities in the bound area ll empirically estimatable" great, but what about your proof tells us that?

Relation to Prior Work: After saying you're not doing verification or falsification, it's confusing to spend so much time in the related work on them. It would be much more informative to discuss alternative estimation approaches, which are almost not reviewed at all. Two specific areas come to mind: 1) Anomaly detection: No work on anomaly detection is surveyed or even mentioned. I am not extremely familiar with the literature, but from a quick google I found many results using density estimation for anomaly detection (including in an RL context): https://www.ukras.org/publications/ras-proceedings/UKRAS20/pp154-156 https://onlinelibrary.wiley.com/doi/full/10.1002/sta4.252 http://www.ifaamas.org/Proceedings/aamas2020/pdfs/p105.pdf http://www.robots.ox.ac.uk/~reece/publications/ICDM12.pdf and also several using neural methods https://arxiv.org/abs/2001.04990 https://www.andrew.cmu.edu/user/lakoglu/odd/accepted_papers/ODD_v50_paper_19.pdf https://ieeexplore.ieee.org/abstract/document/6679866 http://www.doiserbia.nb.rs/Article.aspx?id=1820-02141400035A#.Xyce5ChKiMo https://ieeexplore.ieee.org/abstract/document/8438865 2) inference via neural density estimation e.g. https://openreview.net/pdf?id=S1fcY-Z0- https://arxiv.org/pdf/1804.00779.pdf https://arxiv.org/abs/1810.01367 I'm again not extremely familiar with this literature, so there could be something I've significantly misunderstood, but I think literature on neural autoregressive flows should at least be mentioned, and I wouldn't be surprised if this is very similar to methods that that community uses (e.g. the proposed method seems very similar to using a bayesian hypernetwork with a non-gaussian prior) OOD generalization/sensitivity is measured, but neural methods for OOD generalization, and how this method relates to them, are also not mentioned.

Reproducibility: No

Additional Feedback: Related work and "accessibility" to an ML audience are the biggest weaknesses of this paper in my opinion. I think even if the methods turn out not to benovel compared to existing work, framing them and applying them to safety-critical problems is enough of a contribution to make this an important work and I think that that is worth doing. The method is explained very clearly in terms of how it relates to/combines bridge sampling, neural warping, and adaptive intermediate distributions, and if this included a discussion of neural density / flow methods which most neurips audience would have heard of, I think this would also be a valuable contribution (both for safety and for more general OOD generalization literature) I would be very happy to upgrade my score if these things are done well because I think this type of work is important and interesting.


Review 3

Summary and Contributions: The paper presents a novel sampling paradigm ("Neural Bridge Sampling") that allows to evaluate safety critical systems more quickly by focusing on sampling more rare case events that potentially lead to unsafe behavior. This in turn allows to quickly find tight bounds and thus safety guarantees through simulation, for system that are difficult to formally verify, such as neural networks. The mathematical framework is mathematically sound and thoroughly laid out in the paper and the effectiveness is demonstrated on three interesting and relevant examples. Update: The author feedback adequately addressed my questions regarding putting their work more into context of other sampling techniques. The main point that came up in the reviewer discussion was the ReLU non-reversibility problem. I hope that the authors can adequately address this point in the final revision and give them the benefit of the doubt. I still like the paper and hope it will trigger broader interest in the community. I will keep my original score of 8.

Strengths: This paper presents a solid contribution in the very relevant are of safety verification of complex systems. There is a strong demand for solutions like this, as increasingly complex systems are being deployed in human spaces and formal verification (which was traditionally required for guarantee safety of those systems) is practically not possible anymore. - Solid mathematical formulation - Well structured and line of argumentation well laid out - Method showcased in three different domains - Outperforms MC method in terms of sampling efficiency and ability to compare different systems in terms of safety bounds

Weaknesses: In general, improving the sampling efficiency over naive MC is not new. It would maybe be good to put "Neural Bridge Sampling" into perspective to other sampling approaches in the related work section or even in the evaluation. Disclaimer: I am personally not an expert in sampling techniques.

Correctness: I did not find a flaw in the claims and empirical methodology.

Clarity: Yes, it is well structured and easy to follow. It is a bit equation heavy, but it looks to me like all details are necessary to describe the methods and allow reproducibility.

Relation to Prior Work: Generally the discussion between verification, falsification, and estimation in the related work section is very nice. Again the only point would be to maybe compare it more to other existing sampling methods, and not only naive MC.

Reproducibility: Yes

Additional Feedback:


Review 4

Summary and Contributions: The work presents a novel approach to rare events sampling based on bridge sampling, oriented towards the sampling of adverse events expressed as a probability of a certain function not exceeding a given threshold under a known distribution. The method presented is neural bridge sampling, a combination of (1) bridge sampling for rare events sampling and (2) neural warping that improves Hamiltonian Monte-Carlo sampling. The authors use masked autoregressive flows for warping and a novel adaptive scheme for choosing intermediate distributions. As a result, the authors develop a general approach for estimating probabilities of rare events expressed in the form of quantile functions, probabilities of a certain function not exceeding a certain threshold. This approach is then applied to several different experimental environments, including OpenAI Gym environments and dynamical systems with feedback controls. The authors also provide the code to reproduce their experiments as a supplement. ===== After reading other reviews and the authors' feedback, I still stand by my original positive assessment of the paper. But I would like to point out to the authors that the concerns raised by Reviewer 1 are important, and it will be important to see them addressed in the final version.

Strengths: The authors provide a rigorous analysis of the efficiency of the proposed method, including an estimate on the asymptotic mean-square error which is proven in the appendix. As far as I could check, the proofs are correct and the proposed approach is novel. The experimental section also provides a comprehensive evaluation of the proposed sampling method in several different environments. Overall, I am more than convinced by the experiments. To me, neural bridge warping in the proposed form looks like a great idea that has been convincingly proved to work, both theoretically and experimentally.

Weaknesses: I really liked the paper and the proposed approach but I do not understand why the authors limit the exposition to safety evaluation. This is, of course, the most important application, it should definitely be discussed, and I have nothing against the experimental part being devoted to it, but why put it in the title of the paper and make the entire approach look subordinated to this application? The developed method can be used to estimate small probabilities in the form of basically arbitrary quantile functions, and there are numerous possible applications for it. Therefore, I would advise (but not insist) that the authors give a more general spin to the introduction and put the method (which is their main contribution) in front rather than the application.

Correctness: As far as I could check, the proofs are correct, and the empirical methodology is sound.

Clarity: The paper is well written and easy to follow. It would be nice to put a bit more details from the appendices to the main paper text. In particular, a better explanation of the "Rocket design" case would be great -- right now it's basically impossible to understand what's going on there without consulting the appendix (while the latter is, again, well-written and easy to follow). But I understand that space constraints may overrule this.

Relation to Prior Work: Yes. The work builds on a long history of MCMC-based sampling approaches but the novelty is clearly differentiated. To the best of my knowledge, the proposed algorithm and the main result (Proposition 1) are new.

Reproducibility: No

Additional Feedback:

[Author Response · NeurIPS 2020]

We thank the reviewers for their feedback, and we hope to use the comments to improve our paper. We're glad that the reviewers are enthusiastic about the value of an overall framework and method for efficient rare-event simulation as applied to testing safety-critical systems. In addition to the supplementary code provided, we intend to make our method publicly available and easy to use to garner more impact across the NeurIPS community.

**Related work:** The reviewers' comments mirrored internal discussion we had in writing this draft. We chose to emphasize the application, and, due to space constraints, we focused the literature review on related work in safety testing; we discussed related methods for rare-event sampling only in the context of the method (Section 2). Based on the feedback, however, we will include significant additional discussion of sampling, anomaly detection, and neural density estimation in the revised related work. **R1** notes close relatives of bridge sampling, such as umbrella sampling and multicanonical MCMC. The operational difference between these methods and bridge sampling is in the form of the intermediate distribution used to calculate the ratio of normalizing constants; the optimal umbrella sampling distribution is more brittle than that of bridge sampling. We will also discuss path sampling, a generalization of bridge sampling, wherein we take the discrete bridges to a continuous limit. This is difficult to implement in an adaptive fashion. To answer **R1's** question, we present our approach's relationship to sequential Monte Carlo methods (see e.g. Del Moral, Doucet, Jasra, "Sequential Monte Carlo samplers," 2006) to help readers understand our method. More broadly, our method falls under the formalism of Feynman-Kac models, as do particle filters, birth-death processes, and smoothing filters (see e.g. Del Moral, "Feynman-Kac formulae," 2004). **R1's** reference suggests a slightly narrower view of SMC. Regardless, none of our theoretical results use exogenous machinery or assumptions from SMC.

We also think a broader discussion of rare-event sampling methods (as **R3** suggests) is useful. We will compare non-parametric methods like bridge sampling and multilevel splitting with parametric adaptive importance sampling methods like the cross-entropy method. Our method has both flavors, including parametric warping distributions within the bridge-sampling formalism. Additionally, many of these techniques work with variance-reduction methods like control variates. Recent literature combines importance sampling and kernel methods to make control variate *functions* (e.g. Liu, Lee, "Black-box importance sampling," 2016), which could also provide useful context.

We agree with **R2** that it is useful to include a more thorough discussion on neural density estimation and inference (e.g. Papamakarios et al., "Normalizing Flows [...]," 2019) in the related work. **R2** also asked how our method would work for detection of OOD failures and its relation to the anomaly detection literature. This paper assumes that the generative model of the operating domain $P_0$ is given (lines 30–31), so all failures are in the modeled domain. Therefore, when deploying systems in the real world, anomaly detection (e.g. Nachman and Shih, "Anomaly Detection[...]," 2020, and Choi et al., "WAIC, but Why?[...]," 2019) to discover distribution shifts is complementary to our approach. Another way to frame that problem is via distributional robustness to $P_0$ (Rahimian and Mehrotra, "Distributionally[...]," 2019).

**Hessian computation:** We will clear up confusion **R1** raises regarding the value of eliminating Hessian computation in our approach. Computing $\nabla^2 f$ requires a *double-backward pass through a simulation rollout*. Due to the many time steps involved in the rollout, static computation graphs (e.g. Tensorflow) require exorbitant memory (500 GB of RAM for MountainCar), and dynamic graphs (e.g. PyTorch) simply take a long time to compute. As noted in lines 152–153, Jacobians of flows do not involve $f$, and therefore do not require any backward passes through the simulator. Moreover, the MAF architecture ensures cheap Jacobian computations (in memory/time).

A more fundamental problem is that simulation rollouts are rarely smooth. For example, if the agent's controller contains ReLUs, the map $x \mapsto f(x)$ can be continuous but not smooth. The synthetic experiment showcases the problems that result from using local second-order information from a non-smooth function. We will improve this discussion in the experiments as well as in the introduction (as **R3** suggests).

**Experiments: R1** asks whether our simulators fall into the formalism of nested probabilistic programs: they do not. Specifically, the stochasticity is in initial conditions, and rollouts for given initial conditions are deterministic. We will add a note about the need for other methods (e.g. nested Monte Carlo) for stochastic rollouts. As **R2** suggests, we will illustrate the clustering of failure modes to enhance the experiments. **R2** also raises good points about the connection between the theoretical analysis of NBS's efficiency and the presentation of the experiments. The goal of Eq. (11) is to show that the mean-square error is empirically estimable from a single trial, and we will show this matches the empirical results from repeated experimentation over 10 trials. We will also add discussion around the linear trend of the yellow lines in Figures (1c) and (2c), which matches the theoretical relationship in Proposition 1 and line 215. We will also add additional experiments in the appendix showing performance trends vs. sample size $N$. **R3** suggests comparison with sampling methods beyond naive Monte Carlo. The experiments already include a comparison with adaptive multilevel splitting (AMS) and bridge sampling (B).

**Notation: R1** asked about the "negative ReLU" function, which is $[x]_- := -[-x]_+ = xI\{x < 0\}$; we will clarify this in Section 2. As **R1** and **R2** suggest, we will bring forward some of the discussion about HMC from Appendix A into Section 2 to make the various moving parts of HMC and normalizing flows easier to understand.

**Framing: R4** raised the discussion of framing our paper more generally. Our original intended audience was the safety-testing community, but we will discuss the method's generality in the introduction and conclusion. As **R2** suggests, we will eliminate or define jargon that may be unfamiliar to the broader ML community. Finally, as **R4** suggests, we are considering a title change to broaden the scope of our audience.

[Meta-Review · NeurIPS 2020]

The paper proposes a method for empirical verification of safety system by developing an iterative method for sampling rare and potentially out of bounds system states. The reviewers agree that the strengths are in the novel and important area that is generally little worked in. The method is well founded, rigorously analyzed, and evaluated thoroughly. The methodology presented in the paper is general and applicable to other applications other than the safety, and can generate a lot of follow-up work. In the final version the authors should: - Address R1's concerns about ReLU non-reversibility and show the validity of the bridge-sampling hybrid algorithm empirically or analytically. - Consider positioning the method, and not the application, as the main contribution.